# DiT360: High-Fidelity Panoramic Image Generation via Hybrid Training

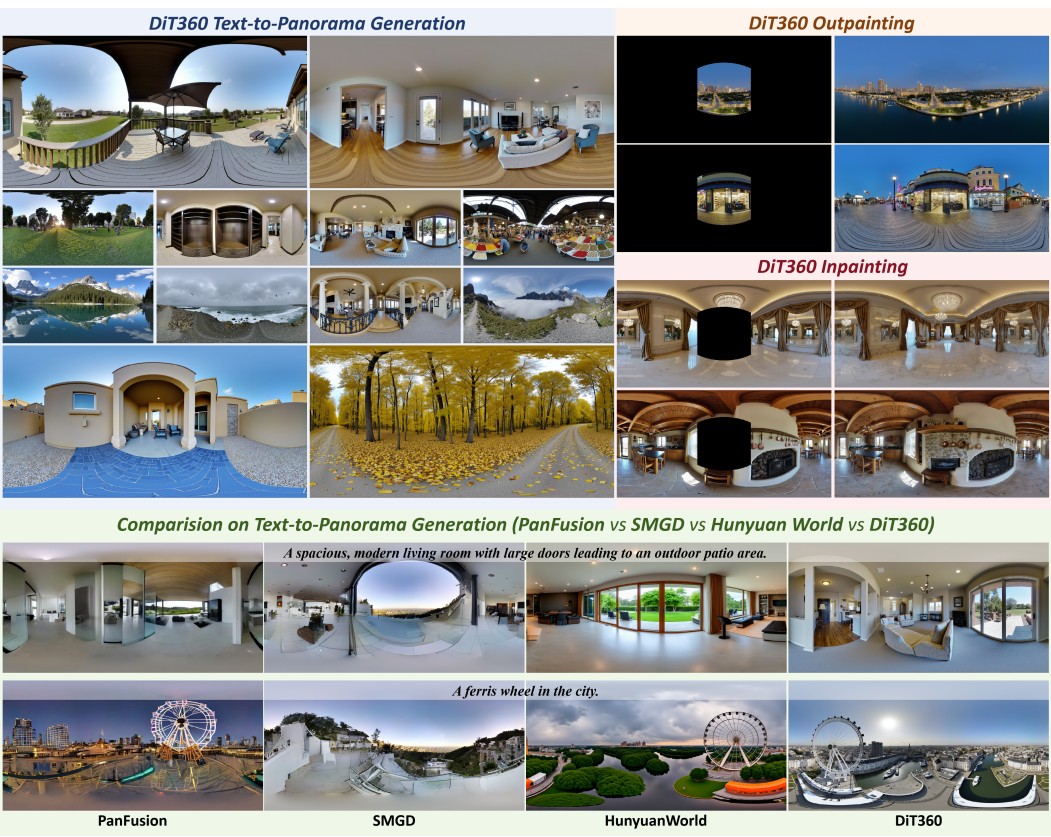

Figure 1: Visualization of *DiT360*'s results. The shown examples include text-to-panorama generation, inpainting, and outpainting, together with comparisons against existing methods.

## Abstract

In this work, we propose *DiT360*, a DiT-based framework that performs hybrid training on perspective and panoramic data for panoramic image generation. For the issues of maintaining geometric fidelity and photorealism in generation quality, we attribute the main reason to the lack of large-scale, high-quality, real-world panoramic data, where such a data-centric view differs from prior methods that focus on model design. Basically, *DiT360* has several key modules for inter-domain transformation and intra-domain augmentation, applied at both the pre-VAE image level and the post-VAE token level. At the image level, we incorporate cross-domain knowledge through perspective image guidance and panoramic refinement, which enhance perceptual quality while regularizing diversity and photorealism. At the token level, hybrid supervision is applied across multiple modules, which include circular padding for boundary continuity, yaw loss for rotational robustness, and cube loss for distortion awareness. Extensive experiments on text-to-panorama, inpainting, and outpainting tasks demonstrate that

our method achieves better boundary consistency and image fidelity across eleven quantitative metrics. Our code, trained models, and curated data will be available.

# 1 INTRODUCTION

With the growing demand for spatial intelligence (Yang et al., 2025a; Chen et al., 2024; Wu et al., 2025), panoramic image generation (Lin et al., 2025) has become critical considering its ability to capture the full 360° field of view. Unlike conventional image generation on perspective views (Esser et al., 2024; Podell et al., 2023; Tian et al., 2024; Black Forest Labs, 2024), panoramic image generation remains challenging due to its unique characteristics such as severe distortions in polar regions, which in turn hinders its wider deployment in applications such as AR/VR and autonomous driving.

To address this issue, existing methods usually focus on specific model design based on equirectangular projection (ERP) (Ye et al., 2024; Huang et al., 2025; Kalischek et al., 2025; Zhang et al., 2024; Xie, 2025; Sun et al., 2025; Team et al., 2025; Ni et al., 2025; Sun et al., 2025), with or without the assistance of cubemaps (CP) (Bar-Tal et al., 2023; Li & Bansal, 2023; Shi et al., 2023; Tang et al., 2023; Park et al., 2025; Yang et al., 2025b), where both ERP and CP are common panoramic representations. Despite the success achieved, these works still struggle with perceptual realism and geometric fidelity due to the scarcity of high-quality real-world panoramic data and the over-reliance on simulated one.

A heuristic solution is to exploit 360° data from media platforms such as YouTube, but its direct use for training is impractical since panoramic data requires domain-specific curation, including horizon correction and aesthetic filtering, which remain largely unexplored. Thus, one question raised: *how can models be endowed with real-world knowledge when only limited panoramic data is available?*

By this motivation, we propose DiT360, a DiT-based framework (Peebles & Xie, 2023), which adopts a hybrid training strategy that combines limited synthetic panoramic data with well-curated, high-quality perspective images to enhance photorealism and geometric fidelity simultaneously. To fully realize the merit of this hybrid paradigm, it is essential to leverage knowledge from the two domains at different representation levels. Accordingly, the *DiT360* incorporates several key modules for inter-domain transformation and intra-domain augmentation, applied at both the pre-VAE image level and the post-VAE token level. At the image level, the focus is on regularization across different domains, where existing panoramic data is regularized through masking and inpainting to remove spatial-variant artifacts in polar regions, while perspective data is regularized into the panoramic space through projection-aware methods to provide photorealistic guidance. At the token level, the focus is on geometry-aware supervision in the latent space. Circular padding aims to address the boundary continuity problem of ERP images, where the left/right edge correspond to inherently periodic 0°/360° longitude. In addition, global rotational consistency is enforced through rotation-consistent yaw loss, while distortion-aware cube loss provides complementary supervision beyond ERP that guides the model toward consistent and high-fidelity panoramic representations.

The extensive experiments demonstrate that *DiT360* with hybrid training can perform better than existing text-to-panorama methods in boundary consistency and image fidelity, as evidenced by both quantitative metrics and qualitative visualizations. For example, DiT360 achieves state-of-the-art performance on the Matterport3D validation set, surpassing prior methods across nine metrics like FID, Inception Score, and BRISQUE. Beyond text-to-panorama generation, *DiT360* naturally supports inpainting and outpainting tasks without additional finetuning enabled by its built-in masking and inpainting strategy. Furthermore, our method can produce high-resolution and photo-realistic panoramic images benefited by high-quality perspective data. Our main contributions are summarized as follows:

- We present *DiT360*, a DiT-based framework with hybrid training that leverages both perspective and panoramic data to preserve photorealism and geometric fidelity. Unlike prior approaches that primarily focus on model design, *DiT360* emphasizes the effective utilization of multi-domain data to achieve superior generation quality.
- The proposed hybrid paradigm is realized through multi-level mechanisms, where image-level regularization refines existing panoramas and leverages perspective data to enhance diversity and photorealism, while token-level supervision in the latent space enforces geometric consistency through rotation- and distortion-aware constraints.

- Extensive quantitative and qualitative experiments on three tasks including text to image, inpainting and outpainting demonstrate that *DiT360* outperforms existing methods in boundary consistency, image fidelity, and overall perceptual quality. The user study conducted further confirms that our method aligns better with human preferences.

## 2 RELATED WORK

**Text-to-Image Diffusion Models.** Diffusion models have replaced earlier approaches (Kingma & Welling, 2022; Goodfellow et al., 2020) as the dominant paradigm in image generation, achieving high-quality and diverse synthesis by reversing a gradual noising process (Dhariwal & Nichol, 2021; Nichol et al., 2022; Saharia et al., 2022; Rombach et al., 2022; Ramesh et al., 2022). Among these, the Latent Diffusion Model (LDM) (Rombach et al., 2022) wrapped with UNet structure introduced denoising in the latent space, enabling scalable high-resolution generation (Podell et al., 2023). More recently, transformer-based architectures (Peebles & Xie, 2023; Vaswani et al., 2017) have been adopted using explicit positional encoding and attention operation to further improve performance (Black Forest Labs, 2024; Esser et al., 2024; Yu et al., 2025; Ma et al., 2024), and are emerging as a new paradigm for better scalability and stronger results. We note that both of UNet- and transformer-based structures are benefited by the large-scale perspective datasets. Motivated by this, we leverage perspective data to compensate for the limited scale of panoramic data by inter-domain transformation and projection.

**Panoramic image Generation.** Early panoramic image generation mainly relied on outpainting-based methods (Akimoto et al., 2022; Dastjerdi et al., 2022; Wang et al., 2022; 2023; Wu et al., 2023b;c; Lu et al., 2024), which reconstruct a full 360° view from partial observations, such as narrow field of view (NFoV), but often suffer from limited flexibility and content diversity. With advances in text-to-image generation, research has shifted towards text-to-panorama generation for more controllable results and can be broadly divided into two categories. The first kinds of approaches (Fang et al., 2023; Höllein et al., 2023; Yu et al., 2023; Bar-Tal et al., 2023; Lee et al., 2023; Li & Bansal, 2023; Shi et al., 2023; Tang et al., 2023; Park et al., 2025; Yang et al., 2025b) generate panoramic images by stitching multiple perspective views. However, they often suffer from limited perceptual realism because of repeated objects and poor geometric fidelity, such as discontinuities. To alleviate this problem, some work (Song et al., 2023; Ye et al., 2024; Huang et al., 2025; Kalischek et al., 2025) adopts cube mapping, which better aligns with the spherical geometry of panoramic images; yet discontinuities across cube faces remain unresolved, along with additional computational and temporal overhead. Another line of work (Chen et al., 2022; Shum et al., 2023; Zhang et al., 2023; Feng et al., 2023; Ai et al., 2024; Wang et al., 2024; Yang et al., 2024a; Zhang et al., 2024; Xie, 2025; Sun et al., 2025; Team et al., 2025; Ni et al., 2025; Wang et al., 2025; Lu et al., 2025) trains models directly on equirectangular images, preserving global continuity and allowing the model to learn distortion patterns. However, these methods struggle to maintain boundary consistency that requires seamless alignment at the 0°/360° longitude and degrade in regions with severe polar distortion, leading to reduced geometric fidelity. Although recent works (Sun et al., 2025; Park et al., 2025; Zhang et al., 2024) attempt to alleviate these issues through alternative convolutional designs, they remain limited in practice and are less compatible with pre-trained models. In addition, all these methods are constrained by the limited quality of panoramic datasets, often inheriting polar degradation and producing rendered-like appearances that lack perceptual realism. In contrast, we employ a hybrid training strategy that enables the generation of high-resolution panoramic images with high perceptual realism, producing sharp and detailed content and strong geometric fidelity, ensuring correct polar distortion and seamless boundaries.

## 3 METHOD

As illustrated in Fig. 2, *DiT360* is a novel framework for generating panoramic images, which improves photorealism and geometric fidelity through hybrid training at both the image and token levels. In the following sections, We first present the preliminaries and overall design of *DiT360* in Sec. 3.1. We then introduce several key modules of the hybrid paradigm from two complementary perspectives: image-level regularization in Sec. 3.2, and token-level supervision in Sec. 3.3. Finally, we show that *DiT360* natively supports extended generation tasks such as inpainting and outpainting without additional training, as detailed in Sec. 3.4.

Figure 2: Overview of the *DiT360* hybrid training pipeline. For the perspective branch, we employ (a) perspective image re-projection to transfer perspective knowledge to panoramic domain. For the panoramic branch, we first apply (b) panoramic refinement to remove polar blurring and then introduce (c) position-aware circular padding, (d) rotation-consistent yaw loss and (e) distortion-aware cube loss for token-level hybrid supervision.

## 3.1 DIT360

**Revisit Diffusion Transformer (DiT).** Recent diffusion models increasingly adopt the DiT architecture (Peebles & Xie, 2023), which uses a transformer (Dosovitskiy, 2020) to process latent sequences of post-VAE image tokens $X \in \mathbb{R}^{N \times d}$, where $N$ is the sequence length and $d$ denotes the embedding dimension. To capture spatial structure, DiT employs Rotary Positional Embeddings (RoPE) (Su et al., 2024), which inject coordinate-dependent rotations into the image tokens, thereby allowing the model to effectively encode both relative and absolute positional information. In addition, DiT adopts a flow-based scheduler to progressively denoise the latent representation, typically conditioned on a text promp $c$. Its training objective is the standard denoising score-matching loss, computed as:

$$\mathcal{L} = \mathbb{E}_{X,c,\epsilon,t}\big[\|\epsilon - \epsilon_\theta(X_t, c, t)\|_2^2\big], \tag{1}$$

where $X_t$ denotes the noise latent in timestep $t$, $\epsilon$ is the added Gaussian noise and $\epsilon_\theta$ represents the predicted noise of the model.

**Overview of *DiT360*.** Building upon DiT, we introduce *DiT360* for panoramic image generation. Fig. 2 illustrates the proposed framework, which adopts a hybrid paradigm to jointly exploit perspective and panoramic data through two training branches. The key modules enabling hybrid training are categorized into **image-level regularization** and **token-level supervision**. At the image level, perspective image guidance and panoramic refinement introduce cross-domain knowledge to enhance perceptual quality while regularizing diversity and photorealism. At the token level, hybrid supervision across multiple objectives is conducted, which includes circular padding for boundary continuity, yaw loss for rotational robustness, and cube loss for distortion awareness. Together, this hybrid design operates across multiple representation levels to achieve perceptual photorealism and geometric fidelity.

## 3.2 IMAGE-LEVEL REGULARIZATION

At the image level, we adopt a hybrid regularization strategy that improves generation quality through inter-domain transformation, combining refinement of existing panoramas with the transfer of photorealistic knowledge from perspective views.

**Panoramic image refinement.** The availability of high-quality, real-world panoramic datasets remains severely restricted, with Matterport3D (Chang et al., 2017) being one of the most widely adopted due to its large scale and high fidelity. Nevertheless, images in this dataset frequently

Figure 3: Panoramic image refinement pipeline. The ERP panorama is converted into a cubemap, where pre-defined masks are applied to the central regions of the top and bottom faces. These masked regions are then reconstructed with an inpainting model and reprojected to ERP. In the figure, orange boxes represent blurry areas, and red dashed boxes indicate inpainted cubes.

exhibit blurring in the polar regions, as shown in Fig. 3, which in turn hampers the quality of downstream panoramic image generation. To mitigate blurring artifacts in the polar regions, we transform panoramic ERPs into cubemap representations, where well-established perspective-domain inpainting can be directly applied. We first fix a binary mask $M$ for each blurred cube face $I$ to localize the inpainting area. Specifically, for $H = W = 1024$, we mask out the central region:

$$M(u,v) = \begin{cases} 0, & \text{if } 256 \leq u, v < 768, \\ 1, & \text{otherwise,} \end{cases} \qquad (2)$$

where $(u, v)$ denotes pixel coordinates. The masked image is then obtained as

$$I_{mask} = I \odot M + (1 - M) \cdot I_{miss}, \qquad (3)$$

where $\odot$ denotes element-wise multiplication and $I_{miss}$ is a white image of the same resolution. Finally, the inpainting model (Labs et al., 2025) is then applied to $I_{mask}$ to reconstruct the missing region and produce $\hat{I}$, which is then transformed back into the ERP space to obtain blur-free, high-fidelity panoramas.. This process serves as an image-quality regularization step, yielding clearer training images while constraining panoramas to retain inherent distortion characteristics, as illustrated in Fig. 3.

**Perspective image guidance.** In addition, we leverage high-quality realistic perspective images from the Internet to regularize the panoramic domain by transferring photorealistic knowledge. Specifically, as illustrated in Fig. 2a, a perspective image is regarded as a cubemap lateral face and then converted back into the ERP representation with a corresponding mask $\mathbf{M}$. We restrict the re-projection to the lateral faces, as the top and bottom faces usually correspond to sky or ground regions, which require perspective images from uncommon viewing angles that are rarely covered in the dataset. During training, we directly apply the mean squre error (MSE) loss from Flux (Black Forest Labs, 2024) to the re-projected ERP, restricting it to the masked regions to avoid contamination from unrelated panoramic areas, yielding:

$$\mathcal{L}_{\text{perspective}} = \mathcal{L}_{\text{MSE}}(\boldsymbol{\epsilon} \odot \mathbf{M}, \hat{\boldsymbol{\epsilon}}_\theta \odot \mathbf{M}), \qquad (4)$$

where $\boldsymbol{\epsilon}$ and $\hat{\boldsymbol{\epsilon}}_\theta$ denote the sampled noise and the reparameterized predicted noise, respectively. This strategy provides effective image-level guidance through cross-domain knowledge adaptation, exposing the model to more diverse scenes and thereby increasing the generation diversity. More importantly, the incorporated perspective knowledge regularizes the model toward photorealistic fidelity, which remains underexplored in prior works.

## 3.3 TOKEN-LEVEL SUPERVISION

At the token level, *DiT360* adopts a hybrid training strategy that balances complementary supervision at the post-VAE token level, simultaneously enhancing boundary continuity, rotational robustness, distortion awareness and perceptual quality. Specifically, we introduce three mechanisms applied to noisy tokens of panoramic inputs: position-aware circular padding for seamless boundary coherence, yaw loss for global rotation consistency, and cube loss for precise supervision of ERP distortion patterns. Together, we propose a hybrid loss design to ensure fine-grained token-level supervision while maintaining balanced generation quality across multiple dimensions.

**Position-aware Circular Padding.**  Panoramic images cover the full 360° horizontal field, making it critical to maintain continuity across image boundaries. To address this challenge, we propose a token-based circular padding mechanism specific to DiT-based frameworks that takes advantage of the inherent correspondence between explicit positional encoding and image content. This property ensures that latent tokens at the same spatial position generate consistent visual features, which we exploit to enhance boundary coherence without introducing additional architectural complexity. Specifically, as illustrated in Fig. 2c, after the VAE compression and the subsequent noise injection, we reshape the latent tokens $X_t \in \mathbb{R}^{N \times d}$ into $X_t \in \mathbb{R}^{H \times W \times d}$. We then apply a circular padding along the width dimension. Formally, let $X_0$ and $X_{-1}$ denote the first and last column features, respectively. We then concatenate them to obtain the padded tensor

$$\tilde{X}_t = \begin{bmatrix} X_{-1}, & X_t, & X_0 \end{bmatrix} \in \mathbb{R}^{H \times (W+2) \times d}. \tag{5}$$

The same operation is applied to the positional encoding, which encourages the model to learn continuity specifically between adjacent columns across the boundary.

**Rotation-consistent Yaw Loss.**  To enforce global rotational robustness, we introduce yaw loss that offers token-level supervision on the model's behavior under spherical rotations along the yaw axis, as illustrated in Fig. 2d. Unlike the standard diffusion loss, the corresponding yaw loss captures the non-linear effects of yaw rotations and constrains the model to produce consistent predictions across different viewing angles. Formally, with the reparameterized noise $\epsilon$ and predicted noise $\epsilon_\theta$, we randomly select a yaw rotation angle $a$ and define the rotated features as:

$$\epsilon_{\text{yaw}} = \text{Rotate}(X_t - \epsilon, a), \quad \epsilon_{\theta, \text{yaw}} = \text{Rotate}(\epsilon_\theta, a), \tag{6}$$

where $\text{Rotate}(\cdot, a)$ denotes the equirectangular panorama rotated by angle $a$ along the yaw axis.

The yaw loss is then computed as the mean squared error (MSE) between the predicted and target rotated noise features:

$$\mathcal{L}_{\text{yaw}} = \mathbb{E}\left[ |\epsilon_{\theta, \text{yaw}} - \epsilon_{\text{yaw}}|_2^2 \right]. \tag{7}$$

**Distortion-aware Cube Loss.**  To effectively model distortion patterns and preserve fine details, we introduce a cube loss based on the cubemap representation of panoramasas, as shown in Fig. 2e. Direct supervision on equirectangular projections often causes the model to reproduce similar distorted appearances rather than learn the precise structural patterns, which leads to incorrect generation details with dealing with polar-region distortions. To address this challenge, we project both sampled and predicted noise onto cube faces and apply face-wise supervision, thereby transferring model's strength in perspective priors to the panoramic domain to preserve structural distortion patterns. Further analysis are provided in appendix A. Formally, let $\mathbf{X}_t$ denote the noisy latent at time step $t$ in the forward diffusion process, $\epsilon$ denote the reparameterized Gaussian noise, and $\epsilon_\theta$ denote the noise predicted by the model. We define the cube-space features by applying a cube-mapping operation:

$$\epsilon_{\text{cube}} = \text{CubeMap}(X_t - \epsilon), \quad \epsilon_{\theta, \text{cube}} = \text{CubeMap}(\epsilon_\theta), \tag{8}$$

where $\text{CubeMap}(\cdot)$ transforms an equirectangular panorama into six cube faces. Then, the cube loss is computed as the MSE between the predicted and target cube-space noise features:

$$\mathcal{L}_{\text{cube}} = \mathbb{E}\left[ |\epsilon_{\theta, \text{cube}} - \epsilon_{\text{cube}}|_2^2 \right]. \tag{9}$$

It is worth noting that we apply both cube and yaw losses directly in the latent noise space. While a natural alternative is to compute them in the latent token space—by predicting latents from noise and comparing with ground-truth latents—our experiments show that noise predictions already encode rich spatial and structural information due to the coupling of noise and semantics in the diffusion objective, making them suitable for spatial supervision. In addition, operating in the noise space aligns the auxiliary losses with the flow-based scheduler, thereby improving training stability.

**Hybrid Loss Design.**  To better balance geometric fidelity and perceptual quality, we adopt a hybrid loss design that retains the MSE loss from Flux (Black Forest Labs, 2024) as the principal objective and augment it with yaw loss and cube loss described above. The overall training loss $\mathcal{L}_{\text{pano}}$ for the panoramic branch is then calculated as:

$$\mathcal{L}_{\text{pano}} = \mathcal{L}_{\text{MSE}} + \lambda_1 \mathcal{L}_{\text{cube}} + \lambda_2 \mathcal{L}_{\text{yaw}}, \tag{10}$$

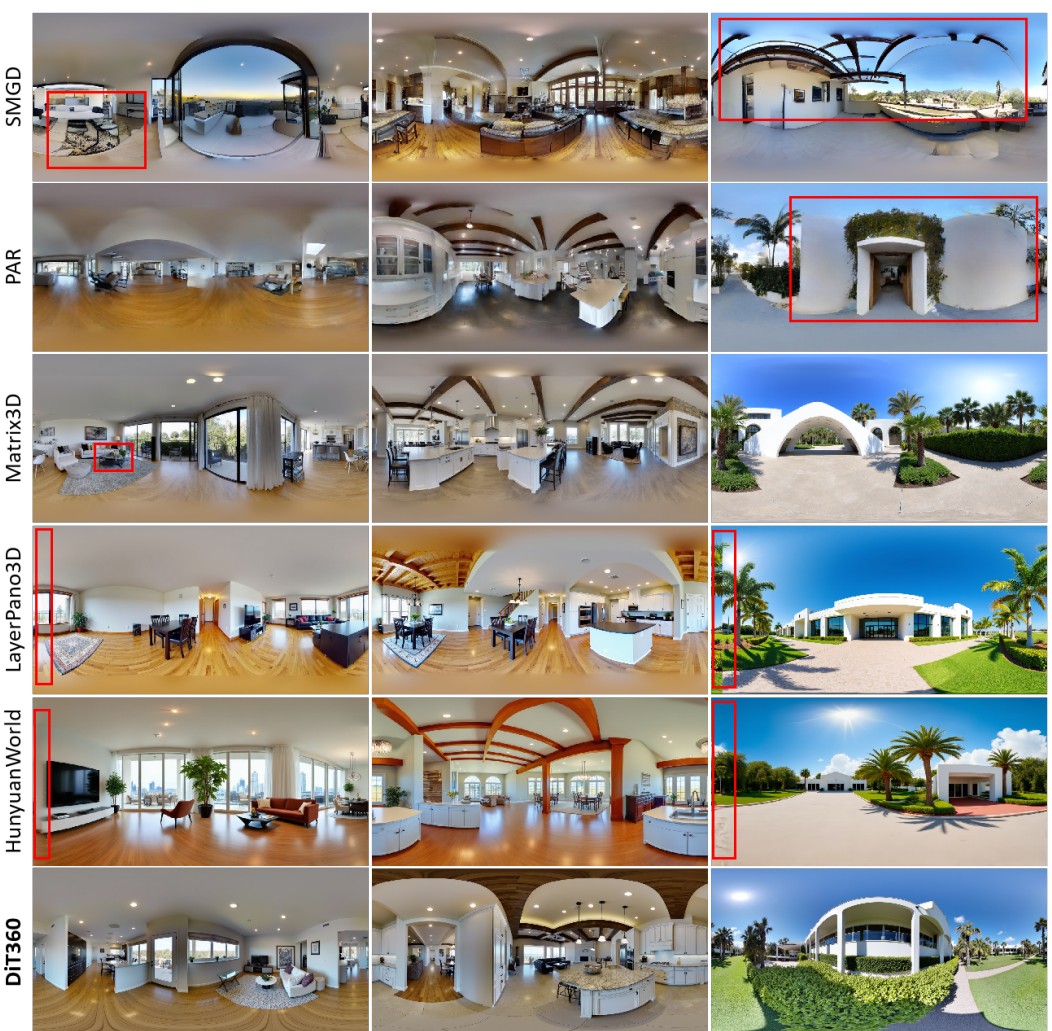

Figure 4: Qualitative comparisons on panorama generation. The representative artifacts are highlighted with red boxes. More complete results are provided in appendix D.

where $\lambda 1$ and $\lambda_2$ represent the balancing coefficients. This token-level hybrid supervision ensures that general perceptual quality, rotational robustness and distortion fidelity are jointly preserved within a unified training framework.

### 3.4 MORE APPLICATIONS

Benefiting from the strong robustness of our method, we perform feature replacement via inversion (Feng et al., 2025) to enable image inpainting and outpainting without additional training. In addition, our model natively supports high-resolution generation, with all training conducted at 1024×2048 resolution. The results in Fig. 1 demonstrate the generalization capability of our approach beyond the primary generation task, with more analysis and results provided in appendix B.

## 4 EXPERIMENTS

### 4.1 SETUP

*DiT360* is developed on top of Flux (Black Forest Labs, 2024) with LoRA (Hu et al., 2021) incorporated into the attention layers. For improved perceptual realism and geometric fidelity, we design

Figure 5: Ablation results of different settings. Artifacts are marked by color-coded bounding boxes: red for spurious details, yellow for boundary discontinuities, and green for incorrect distortions.

Table 1: Quantitative comparison results on text-to-panorama generation. We highlight the best result in **bold** and the second best with underline.

| Methods | FID↓ | FID$_{clip}$↓ | FID$_{pole}$↓ | FID$_{equ}$↓ | FAED↓ | IS↑ | CS↑ | QA$_{quality}$↑ | QA$_{aesthetic}$↑ | BRISQUE↓ | NIQE↓ |
|---|---|---|---|---|---|---|---|---|---|---|---|
| PanFusion | 124.87 | 120.75 | 182.09 | 108.12 | 11.06 | 1.30 | 28.35 | 3.83 | 3.56 | 27.38 | 4.31 |
| MVDiffusion | 108.19 | 117.26 | - | - | 4.39 | 1.58 | 34.65 | 3.97 | 3.25 | 44.79 | 4.91 |
| SMGD | 46.72 | 45.04 | 65.69 | 34.84 | 3.29 | 1.40 | 31.14 | 4.05 | 3.77 | 30.35 | 4.75 |
| PAR | 47.72 | 47.26 | 76.93 | 27.39 | 2.97 | 1.34 | 33.85 | 3.91 | 3.54 | 32.26 | 4.38 |
| WorldGen | 67.11 | 62.97 | 79.32 | 33.45 | 3.29 | 1.40 | 34.61 | 4.30 | 3.59 | 32.31 | 4.82 |
| Matrix-3D | 60.91 | 56.70 | 77.21 | 26.73 | 3.08 | 1.56 | 34.59 | 4.48 | 3.78 | 16.37 | 3.95 |
| LayerPano3D | 62.82 | 60.34 | 80.37 | 38.67 | 2.98 | 1.50 | 34.40 | **4.73** | 3.93 | 33.91 | 3.79 |
| HunyuanWorld | 76.75 | 75.65 | 106.58 | 41.75 | **2.91** | 1.53 | **34.73** | 4.67 | 3.85 | 39.12 | 5.18 |
| **Ours** | **42.88** | **41.60** | **50.88** | **24.77** | **2.91** | **1.60** | 34.68 | 4.69 | **4.19** | **10.25** | **3.72** |

a hybrid training strategy that combines high-quality Internet landscape images with large-scale panoramas from Matterport3D (Chang et al., 2017). To assess the effectiveness of our approach, we adopt a diverse set of complementary metrics covering realism, diversity, text–image alignment, and perceptual quality, ensuring a comprehensive assessment of model performance. More detailed descriptions of the implementation, dataset preprocessing, and metric definitions are in appendix C.

## 4.2 Main Results and Comparisons

In this section, we present our main experimental results and conduct a comprehensive comparison with representative baselines, including PanFusion (Zhang et al., 2024), MVDiffusion (Tang et al., 2023), SMGD (Sun et al., 2025), PAR (Wang et al., 2025), WorldGen (Xie, 2025), Matrix-3D (Lu et al., 2025), LayerPano3D (Yang et al., 2024b), and HunyuanWorld (Team et al., 2025). These methods span diverse architectures, enabling a comprehensive evaluation of our approach. In addition to quantitative and qualitative comparisons, we also present a user study in appendix E and further results in appendix F.

**Qualitative Comparisons.** We provide qualitative comparisons with baseline methods in Fig. 4 and highlights artifacts with red boxes. SMGD (Sun et al., 2025) and PAR (Wang et al., 2025) propose alternative paradigms—structural modifications or autoregression—but struggle with detail fidelity, often producing cluttered or less precise results. Moreover, insufficient data quality leads to pronounced distortions near the polar regions, resulting in poor perceptual realism. Recent advances in Diffusion Transformers (DiT) (Peebles & Xie, 2023) have led to their adoption as backbones in several panorama generation methods. Matrix-3D improves boundary alignment yet struggles with fine-grained details, suffering from limited geometric fidelity. LayerPano3D (Yang et al., 2024b) and HunyuanWorld (Team et al., 2025) leverage large amounts of synthetic data, which improves geometric fidelity to some extent, but results in render-like appearances that compromise perceptual realism; additionally, iterative denoising introduces further artifacts. In contrast, our method generates panoramas with high perceptual realism and geometric fidelity, producing sharp, detail-preserving images with strong lateral consistency and effectively mitigated distortions.

Table 2: Ablation study of different model components on text-to-panorama generation. Best results are in **bold**, second best are underlined, and "Pi guidance" denotes perspective image guidance.

| Methods | FID↓ | FID$_{clip}$↓ | FID$_{pole}$↓ | FID$_{equ}$↓ | FAED↓ | IS↑ | CS↑ | QA$_{quality}$↑ | QA$_{aesthetic}$↑ | BRISQUE↓ | NIQE↓ |
|---|---|---|---|---|---|---|---|---|---|---|---|
| Flux + LoRA | 46.69 | 45.90 | 66.03 | 28.91 | 3.23 | 1.51 | 34.39 | 4.40 | 3.97 | 17.02 | 3.97 |
| w/ circular padding | **43.71** | **42.36** | 61.32 | 27.51 | 3.04 | 1.54 | 34.44 | 4.51 | 3.98 | **13.61** | **3.82** |
| w/ cube loss | 44.40 | 43.75 | **60.16** | **26.30** | 3.01 | **1.57** | **34.62** | 4.41 | 3.92 | 15.68 | 3.89 |
| w/ yaw loss | 44.63 | 43.90 | 64.19 | 26.99 | **2.98** | 1.56 | 34.53 | 4.37 | 3.94 | 15.96 | 3.92 |
| w/ pi guidance | 46.03 | 44.92 | 63.72 | 27.81 | 2.95 | 1.48 | 34.42 | **4.54** | **4.02** | 16.94 | 3.83 |
| **Ours (w/ all)** | 42.88 | 41.60 | 50.88 | 24.77 | 2.91 | 1.60 | 34.68 | 4.69 | 4.19 | 10.25 | 3.72 |

**Quantitative Comparisons.** We further conduct quantitative evaluations to validate the effectiveness of our approach, with results reported in Tab. 1. Our method ranks first on nearly all benchmarks and shows consistently strong performance across most metrics. Although our approach slightly underperforms the top methods on CLIP Score and the quality branch of Q-Align, the gaps are marginal and largely attributable to the fact that both metrics are designed for perspective images, which may not fully reflect the quality and fidelity of panoramas. Collectively, the results support our qualitative observations and demonstrate the effectiveness and robustness of our approach in generating high-quality panoramas.

## 4.3 ABLATION STUDY

To assess the contribution of each component, we conduct ablation studies using a combination of Flux (Black Forest Labs, 2024) and LoRA (Hu et al., 2021) as the baseline. We ablate four key modules: position-sensitive circular padding, distortion-sensitive cube loss, rotation-consistent yaw loss, and perspective image guidance and evaluate their impact in Tab. 2 and Fig. 5.

**Circular padding** significantly enhances consistency across image boundaries and also improves overall image quality, reflected in reductions of FID (Heusel et al., 2018) and BRISQUE (Mittal et al., 2012), because the identical positional encoding on the left and right edges allows the model to learn correct boundary correspondences.

**Cube loss** mainly refines fine-grained details and reduces artifacts by applying additional supervision on the cubemap representation, enabling the model to learn accurate panoramic distortions. This results in substantially fewer artifacts in the polar regions and thus largely improved IS and CS that are more related to the visual semantics.

**Yaw loss** improves global rotation consistency and structural coherence, explaining its superior performance on FAED (Oh et al., 2021) where the autoencoders used are pre-trained by panoramic images. This is because that we supervise the model on rotated tokens to explicitly enforce full-image rotation consistency.

**Perspective image guidance** further enhances local details, enriches visual diversity and effectively mitigates detail-related artifacts, as evidenced in the QA$_{quality}$ and QA$_{aesthetic}$ metrics that are more sensitive to the visual style.

Overall, the components contribute to the perceptual realism and geometric fidelity, and their combination delivers the strongest performance, validating the effectiveness of our framework.

## 5 CONCLUSION

In this paper, we proposed *DiT360*, a framework for geometry-aware and photorealistic panoramic image generation, built upon a hybrid training strategy that combines limited high-quality panoramic data with large-scale perspective images to enhance both realism and generalization. To fully leverage this hybrid paradigm, we introduce multiple modules across different representation levels, where image-level regularization refines existing panoramas and leverages perspective data to enhance diversity and photorealism, while token-level supervision in the latent space enforces geometric consistency through rotation- and distortion-aware constraints. Extensive experiments on text-to-panorama generation, inpainting, and outpainting demonstrate superior image fidelity, boundary consistency, and visual quality. By bridging perspective and panoramic domains across multiple representation levels, *DiT360* establishes a strong baseline for future research in 3D scene generation and large-scale open-world environments.

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

# APPENDIX

## A    EFFECT OF SUPERVISION ON POLAR DISTORTIONS

In this section, we further illustrate the effect of cube loss in addressing severe distortions around the polar regions. Figure 6 compares results generated from the same prompt without and with this supervision, showing that incorporating cube loss leads to clearer structures and fewer artifacts in the polar regions.

*A modern bathroom with a toilet and a window with shutters.*

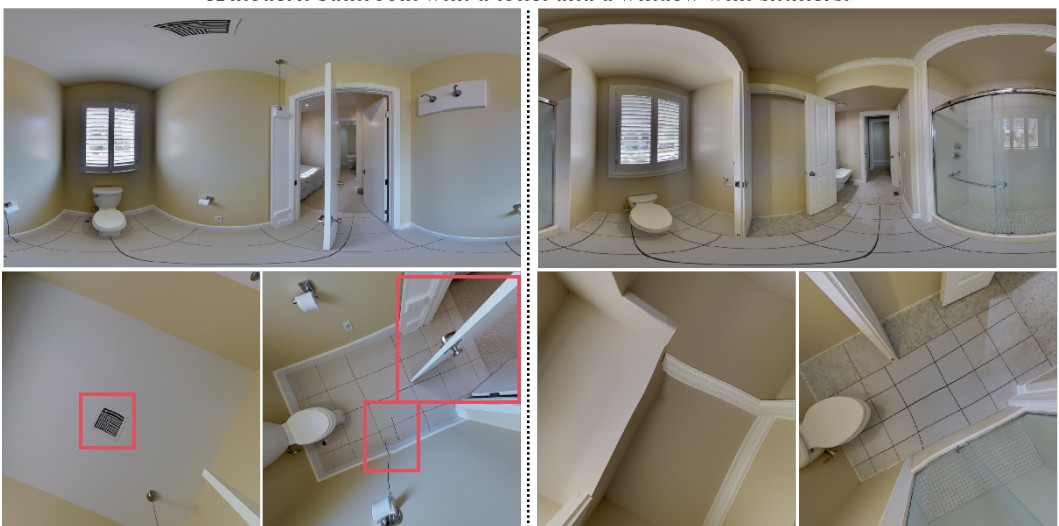

Figure 6: Qualitative comparison of generated panoramas and their top/bottom cube faces without (left) and with (right) cube loss. Red boxes mark regions where polar artifacts are significantly reduced when supervision is applied.

## B    INPAINTING AND OUTPAINTING

*DiT360* demonstrates native inpainting and outpainting capabilities without requiring additional training, thereby establishing a unified framework for panoramic image generation. Specifically, inspired by (Feng et al., 2025), we first perform inversion on the input image to obtain its initial noise representation. At the same time, we extract reference image tokens without positional encodings, along with the associated subject mask. During the early denoising steps, we employ a token replacement strategy. The tokens within the masked or extended regions are substituted with those from the reference image, while preserving the original positional encodings. This time-step-adaptive replacement mechanism ensures faithful reproduction of subject details and spatial consistency. It anchors subject identity in the early phase of generation and naturally guides the model toward coherent content completion. As a result, *DiT360* produces consistent and semantically rich results in both inpainting and outpainting tasks. More results are provided in Fig. 7.

## C    EXPERIMENT SETTINGS

**Implementation Details.**    We developed *DiT360* on top of Flux (Black Forest Labs, 2024), integrating LoRA (Hu et al., 2021) into the attention layers. The model was fine-tuned on 5 H20 GPUs using AdamW (Loshchilov & Hutter, 2019) with a learning rate of $2 \times 10^{-5}$ for 20 epochs, a batch size per GPU of 1, and a gradient accumulation of 3. Our experiments revealed that the guidance scale plays a crucial role in convergence, with 1.0 yielding the most stable training. For inference, we set the guidance scale to 3.0 and employed 28 sampling steps.

*A panoramic view of city skyline from the river.*

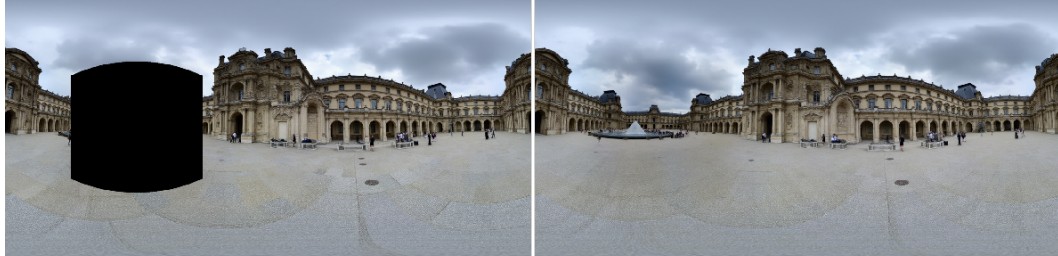

*Florence central plaza with historic buildings and fountains.*

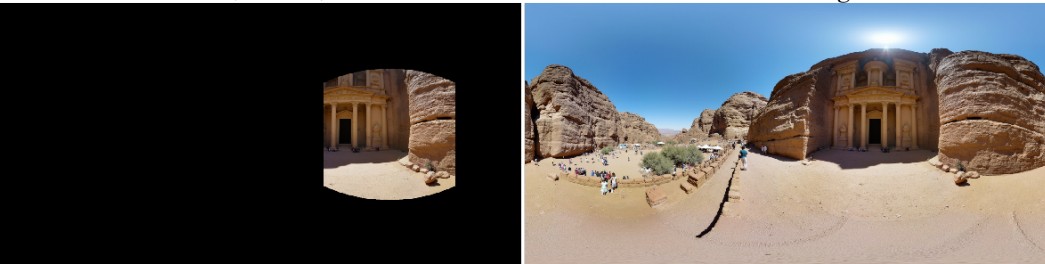

*Petra, Jordan, with rock-cut architecture and desert surroundings.*

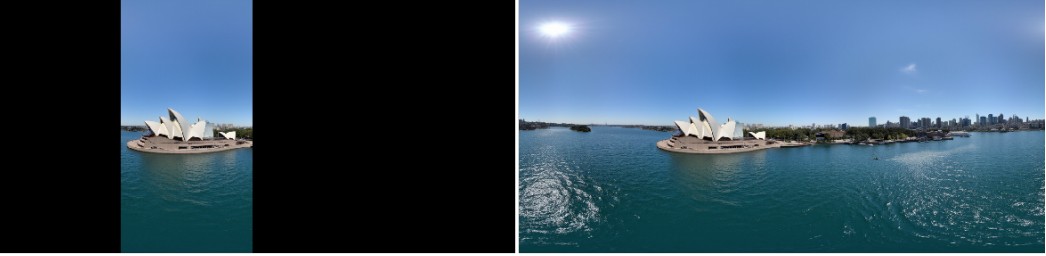

*Sydney Opera House with harbor and skyscrapers.*

Figure 7: More results on inpainting and outpainting.

**Dataset.** We adopt a hybrid training strategy that combines perspective and panoramic data. For the perspective branch, we curate 40k high-quality landscape images from the Internet, center-crop them to a 1:1 ratio, and project them onto random panoramic regions. For the panoramic branch, we follow PanFusion (Zhang et al., 2024) and utilize Matterport3D (Chang et al., 2017), a large-scale RGB-D dataset comprising 10,800 panoramas across 90 building-scale scenes. To mitigate distortion, we refine the blurred polar regions and use 10k panoramas for training while reserving the remainder for validation, consistent with prior work.

**Evaluation Metrics.** Following prior work, we evaluate our method with a diverse set of complementary metrics. For realism, we adopt Fréchet Inception Distance (FID) (Heusel et al., 2018) and its variants, including $FID_{clip}$ for fair comparison by excluding blurred polar regions, and $FID_{pole}$ and $FID_{equ}$ following SMGD (Sun et al., 2025) to assess polar distortion and perspective projection quality. Since FID relies on an Inception network trained on perspective images and may not fully capture panoramic characteristics, we further employ Fréchet Auto-Encoder Distance (FAED) (Oh

et al., 2021), a variant tailored for panoramas. For diversity, we report Inception Score (IS) (Salimans et al., 2016), replacing the standard Inception-v3 (Szegedy et al., 2015) with a ResNet pretrained on Places365 (He et al., 2015; Zhou et al., 2017) to better reflect the scene-centric nature of our data. For text–image alignment, we compute CLIP Score (CS) (Radford et al., 2021), and for perceptual quality, we report Q-Align (QA) (Wu et al., 2023a), BRISQUE (Mittal et al., 2012), and NIQE (Mittal et al., 2013), following HunyuanWorld (Team et al., 2025).

## D  FULL COMPARISION

In this section, we present the complete qualitative comparison of text-to-panorama generation results. As shown in Fig. 8, our method demonstrates superior perceptual realism, producing sharper and more visually authentic panoramas. In addition, it achieves higher geometric fidelity by effectively handling distortions and preserving boundary continuity, whereas baseline methods often suffer from visible artifacts and structural inconsistencies.

## E  USER STUDY

Table 3: User study results on text-to-panorama generation.

| Methods | Text Alignment↑ | Boundary Continuity↑ | Realism↑ | Overall Quality↑ |
|---|---|---|---|---|
| PanFusion | 21.7% | 19.6% | 2.1% | 0.3% |
| Matrix-3D | 24.1% | 27.5% | 23.7% | 5.1% |
| HunyuanWorld | 25.9% | 18.9% | 10.4% | 13.7% |
| Ours | **28.3%** | **34.0%** | **63.8%** | **80.9%** |

To further evaluate human preference, we conducted a user study comparing our method with several representative baselines (Zhang et al., 2024; Lu et al., 2025; Team et al., 2025). The study focused on four key aspects: text alignment, boundary continuity, realism, and overall quality. A total of 63 participants were asked to choose their preferred outputs from different methods on the test set. As shown in Tab. 3, our method received the highest preference across all metrics, clearly demonstrating its superior ability to generate realistic panoramic images with faithful alignment and coherent boundaries.

## F  MORE RESULTS

We present additional results in Figs. 9 and 10 to further illustrate the performance of *DiT360* on panoramic image generation. These examples demonstrate that the model consistently produces high-quality, semantically coherent, and visually detailed completions across a variety of scenes.

## G  USE OF LARGE LANGUAGE MODELS

Large Language Models were used for minor grammar and style corrections only. All technical content, experiments, and conclusions were authored by the paper's authors.

## H  LIMITATIONS AND FUTURE WORK

Despite the strong performance of *DiT360* on panoramic image generation tasks, several limitations remain. The model's effectiveness is constrained by the diversity and scale of available datasets, leading to suboptimal results in certain scenarios, such as those containing high-resolution human faces or intricate scene details. Future work will focus on collecting larger and more diverse high-quality datasets to further enhance the model's generative capabilities and image resolution. Additionally, leveraging synthetic data to augment training samples can facilitate further advances in panoramic image generation. In the long term, extending the framework to three-dimensional scene generation and understanding represents a promising research direction.

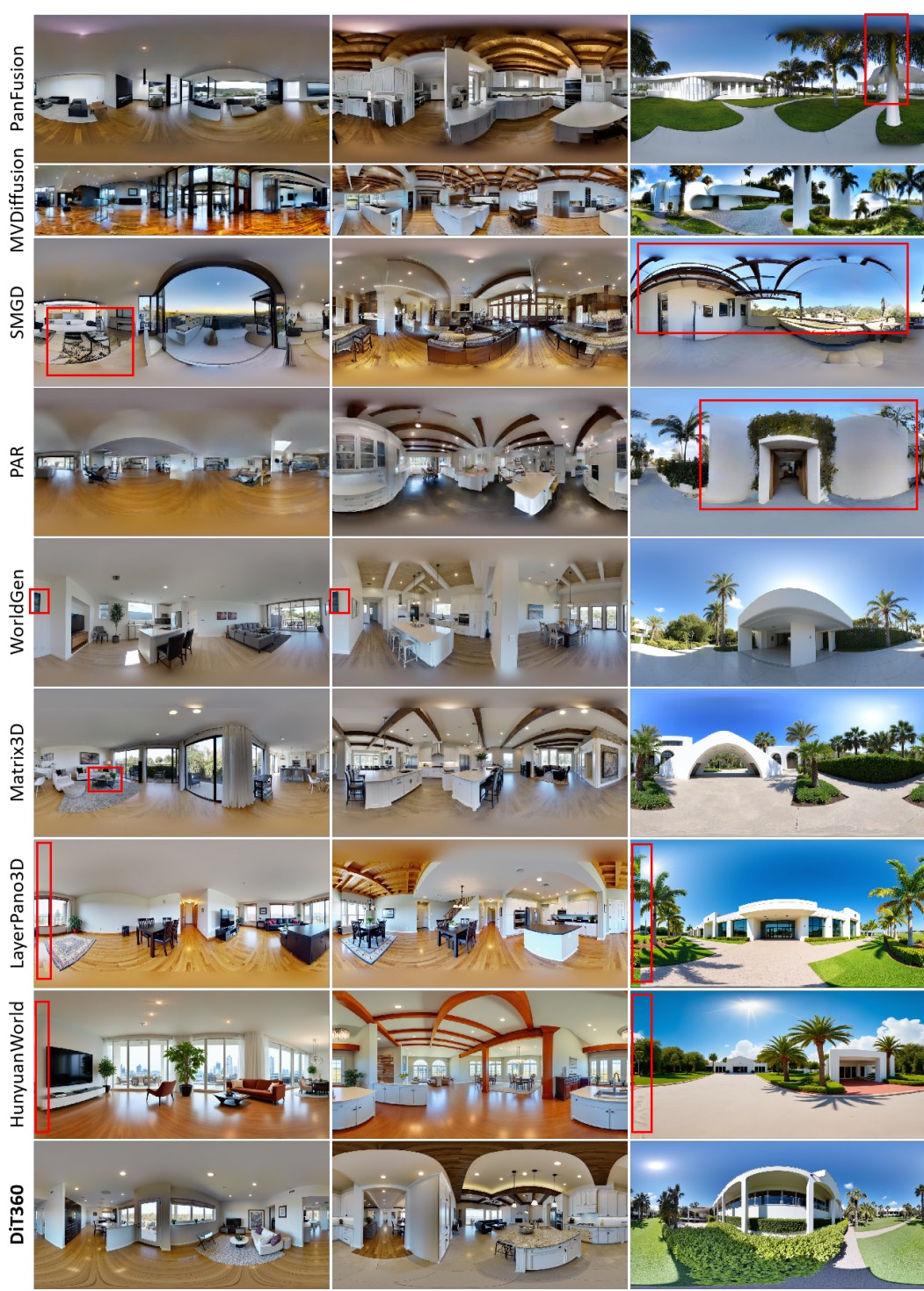

Figure 8: The full qualitative comparison on panorama generation. We highlight representative artifacts with red boxes.

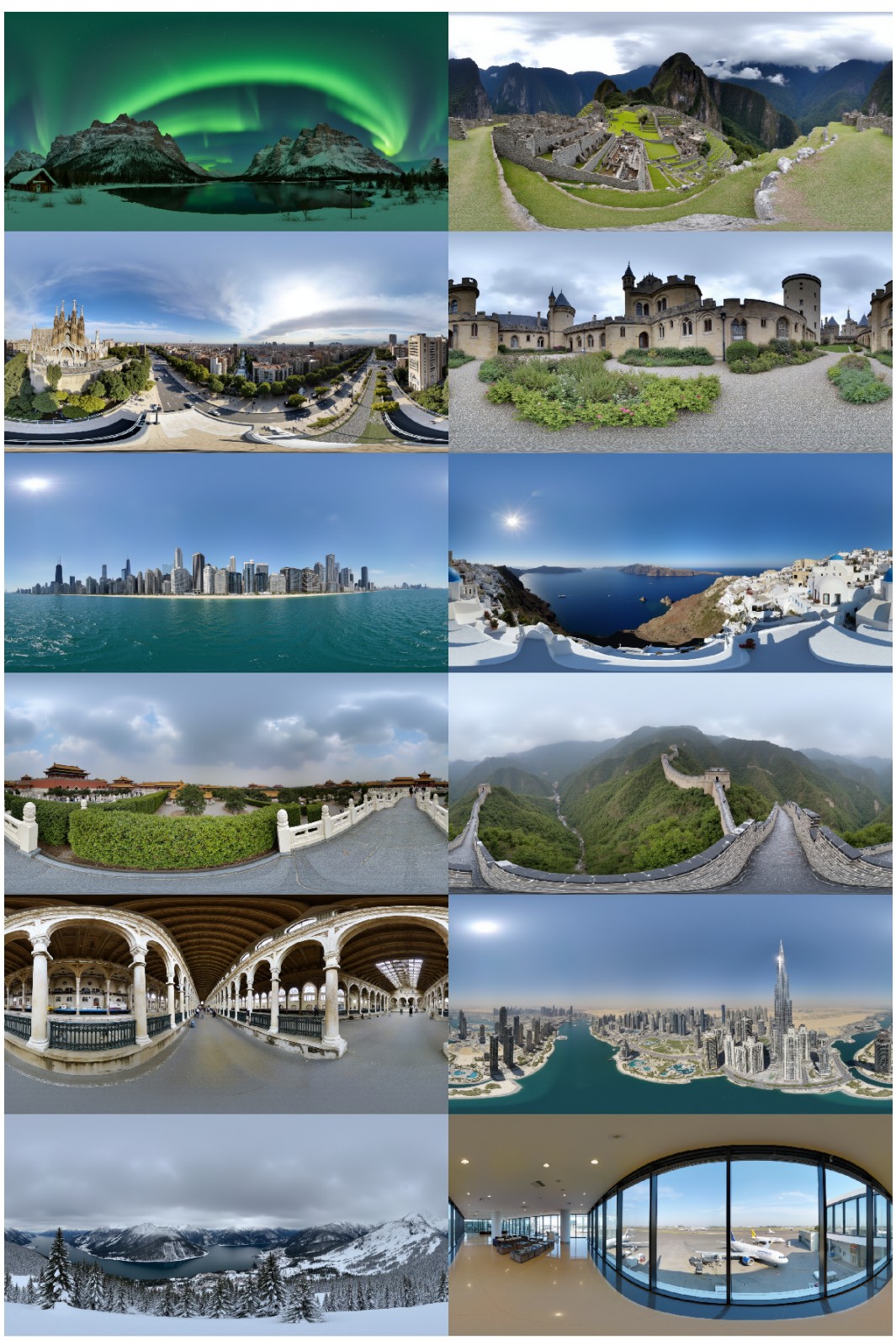

Figure 9: More results on text-to-panorama generation.

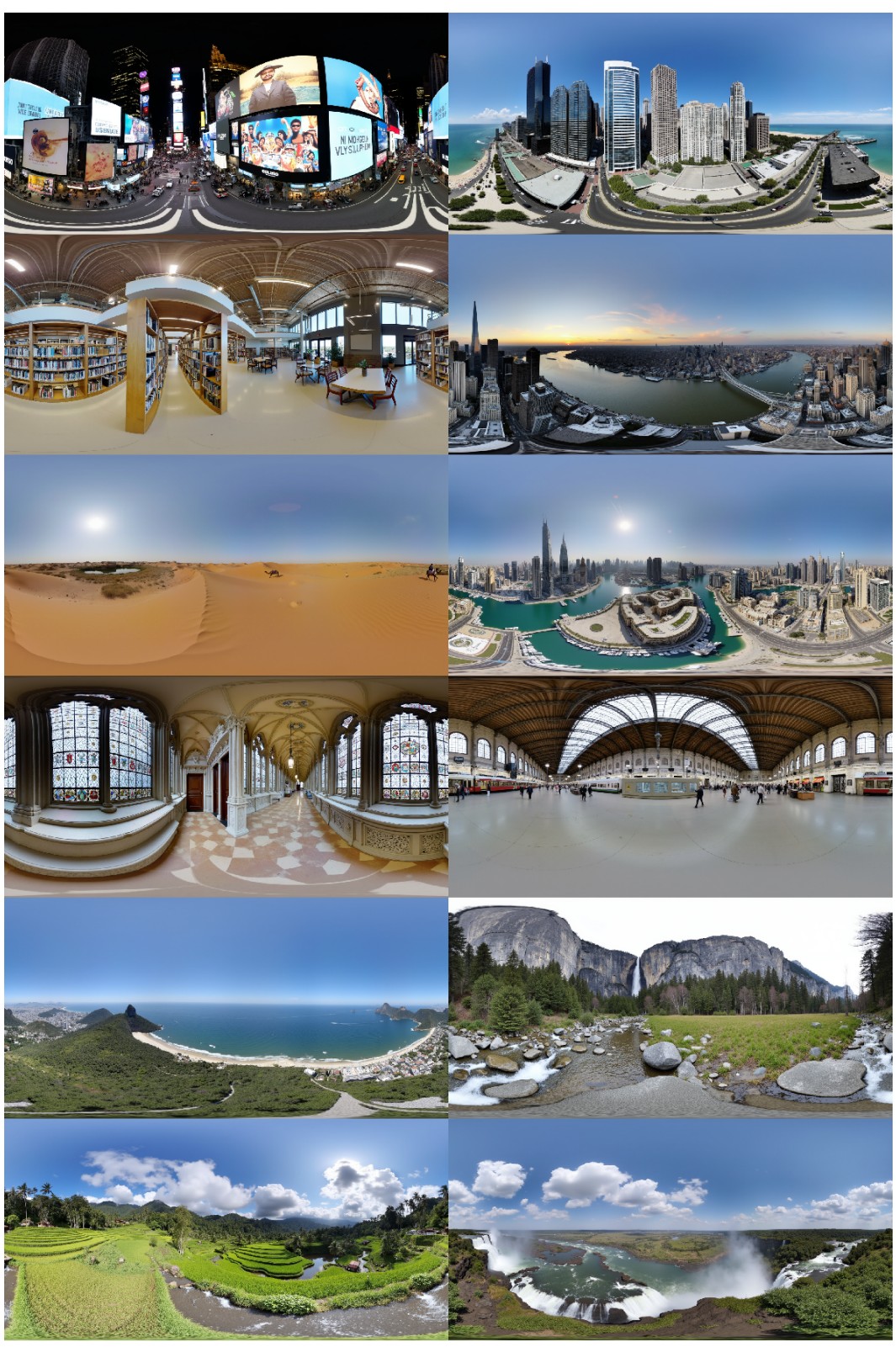

Figure 10: More results on text-to-panorama generation.

