# OpenReview forum: "DiT360: High-Fidelity Panoramic Image Generation via Hybrid Training"
_ICLR.cc/2026/Conference — ICLR 2026 Conference Withdrawn Submission_

### Official Review · Reviewer_Zwpe · 2025-10-24

**Soundness:** 2
**Presentation:** 3
**Contribution:** 2
**Rating:** 2
**Confidence:** 5

**Summary:**

This paper presents a framework for panoramic image generation by leveraging a hybrid training scheme on perspective and panoramic data. It introduces some data augmentation and loss to enhance the model's performance. However, the designs need more justification and evaluation is insufficient.

**Strengths:**

- Extensive experiments have been conducted, and with some extended experiments like inpainting and outpainting task;

- Some interesting loss functions have been proposed.

**Weaknesses:**

- Regarding the "Perspective image guidance" mentioned in the paper, the approach appears to involve simply incorporating certain masks into the loss function. This raises the question of whether the dit model has already been exposed to and computed attention for black regions in data constructed under such conditions. Relying solely on such a mask-based loss is insufficiently justified and does not make sense. More details should be discussed.

- About the propesed "Rotation-consistent Yaw Loss". This loss resembles a form of data augmentation through rotation operations applied in the noise domain, which has been widely applied in previous works. The paper would benefit from a more rigorous theoretical or empirical argument explaining why this form of regularization is particularly effective, especially compared to standard augmentation in image level, and how it fundamentally improves latent space organization or generation quality.

- The "Position-aware Circular Padding" seems a known trick in previous panorama generation paper to improve image boundaries. Any difference here? To better validate this contribution, the authors should include quantitative or qualitative results that explicitly demonstrate the method's performance in handling boundary inconsistency.

- Insufficient evaluation. There is only evaluation for text-to-panorama generation. Why not evaluate image-to-panorama generation? As claimed by authors (and also shown in Fig. 1), the model can perform panorama outpainting, which is exactly the task of image-to-panorama. As many baseline models and the proposed method can perform image-to-panorama gen, more evaluation are needed.

- Which benchmark did you use for quantitative evaluation in table 1? This is not clear and the results of some baselines in table 1 do not make sense. for example, hunyuan world clearly outperforms layerpano3d and worldgen in text-to-panorama in many papers'evaluation. Also lack of CLIP score to evaluate input-output alignment?

- generalization. The visual results shown in paper only involve photorealsitic images. how about stylized images? also, stylized images are used in evaluation in table 1? this is very unclear and unreproducable.

**Questions:**

see above. the evaluation is extremely insufficient with unclear settings.

---

### Official Review · Reviewer_oajz · 2025-10-30

**Soundness:** 3
**Presentation:** 2
**Contribution:** 3
**Rating:** 4
**Confidence:** 5

**Summary:**

The paper presents DiT360, a hybrid training strategy for panoramic image generation using the DiT model. Since panoramic data are scarce and differ from normal image data, training with a pure DiT model results in poor generation quality. To address this issue, the authors introduce image-level regularization and token-level supervision. In image-level regularization, panoramic image refinement inpaints the blurry polar regions in the Matterport3D dataset, and perspective image guidance uses MSE loss on the ERP representation of perspective images during training. In token-level supervision, position-aware circular padding ensures avoidance of seams between the left and right sides of the image, and cube and yaw losses are used to reduce distortion. Experiments demonstrate the effectiveness of the proposed training framework for panoramic image generation, both quantitatively and qualitatively.

**Strengths:**

1. In the case of panoramic image refinement, the performance was improved with solid motivation and a simple approach.

2. Circular padding also has a clear motivation that previous works have not addressed, and it was solved by leveraging the characteristics of Flux.1-dev.

3. Two different types of data were trained to improve performance using learning techniques specific to panoramic images.

4. Each of the proposed modules created synergy, achieving a greater performance improvement compared to basic Flux + LoRA training.

**Weaknesses:**

Overall, the proposed method demonstrates notable performance improvements. However, certain descriptions in the paper convey inaccurate or unclear information, and several key experimental details are missing, which hinders full reproducibility and proper evaluation of the results. I am willing to raise the rating if the items below are resolved:

1. In L185-196, the pure DiT doesn’t have RoPE and doesn’t use flow-based scheduler. If the model uses a flow-based scheduler, Eq. 1 must contain the signal-to-noise ratio $\lambda$ and the time-dependent weighting parameter $\omega$. I think the authors are trying to introduce Flux, but the current writing is confusing. Please specify the model name and describe its characteristics correctly.

2. Although it is not a major part of this paper, I think it is a small contribution to identify defects in the Matterport3D dataset, perform pre-processing (inpainting using Flux.1 Kontext), and improve data quality. Then, did the evaluated baseline models also learn from the pre-processed data for comparison? If not, it does not seem to be a fair comparison with the baseline (Tab. 1).

3. The ablation of panoramic image refinement is missing. Data quality is a very important factor in learning, so please include ablation for this.

4. The paper should explicitly state the exact model version used (I believe it is Flux.1-dev). The current description leaves this ambiguous, forcing readers to infer details that should be clearly specified.

5. Please also clarify the training configuration. Specifically, whether only the additional LoRA layers were trained or if both the base Flux model and the LoRA layers were jointly trained.

6. In L788-789, the authors mention collecting 40K Internet data samples. Since these data are presumably non-panoramic, it is unclear how they are utilized within the proposed framework. Is only the Perspective Branch trained in this case? If so, please clarify the training procedure and its integration with the overall model.

**Questions:**

1. How much time and GPU memory are required to generate a single 1024 $\times$ 2048 image?

**Details Of Ethics Concerns:**

No concern.

---

### Official Review · Reviewer_oktw · 2025-11-01

**Soundness:** 3
**Presentation:** 3
**Contribution:** 3
**Rating:** 8
**Confidence:** 4

**Summary:**

The paper introduces DiT360, a framework for panoramic image generation that leverages hybrid training on both perspective and panoramic data. The method augments a DiT-based diffusion model with several modules—circular padding, yaw loss, and cube loss—to improve geometric fidelity and photorealism. The approach is evaluated on text-to-panorama, inpainting, and outpainting tasks, showing improvements over prior baselines in both quantitative metrics and user studies.

**Strengths:**

- The hybrid training strategy is well-motivated. While not inherently novel, the design and execution of the framework is well done, and the results show this.
- The pipeline is well thought out, with clear ablations showing the impact of each module (this is nicely done). The paper is clear and easy to follow.
- The method demonstrates strong qualitative results, particularly in boundary consistency and reduction of polar artifacts.
- The inclusion of a user study and a broad set of evaluation metrics is appreciated.

**Weaknesses:**

- The generalization to domains beyond indoor panoramas (e.g., outdoor, dynamic scenes) is not demonstrated.
- Lack of ablations/insight into $λ_1$ and $λ_2$
- A primary use case, particularly in AR/VR, is high-frequency details, human faces, etc. This method does not address these areas.

**Questions:**

- How robust is the method to domains outside Matterport3D, such as outdoor or highly dynamic scenes?
- Can the authors provide more quantitative analysis of failure cases, especially for high-frequency details and human faces?
- How sensitive is the method to the choice of loss weights (λ1, λ2) in the hybrid loss design?

---

### Note · Authors · 2025-11-13

I have read and agree with the venue's withdrawal policy on behalf of myself and my co-authors.